# Solving Equations of Motion by Using Monte Carlo Metropolis: Novel Method Via Random Paths Sampling and the Maximum Caliber Principle

**DOI:** 10.3390/e22090916

**Published:** 2020-08-21

**Authors:** Diego González Diaz, Sergio Davis, Sergio Curilef

**Affiliations:** 1Departamento de Física, Universidad Católica del Norte, Casilla 1280, Antofagasta, Chile; scurilef@ucn.cl; 2Banco Itaú-Corpbanca, Casilla 80-D, Santiago, Chile; 3Comisión Chilena de Energía Nuclear, Casilla 188-D, Santiago, Chile; sdavis@cchen.cl

**Keywords:** maximum caliber, Monte Carlo Metropolis, equation of motion, least action principle

## Abstract

A permanent challenge in physics and other disciplines is to solve Euler–Lagrange equations. Thereby, a beneficial investigation is to continue searching for new procedures to perform this task. A novel Monte Carlo Metropolis framework is presented for solving the equations of motion in Lagrangian systems. The implementation lies in sampling the path space with a probability functional obtained by using the maximum caliber principle. Free particle and harmonic oscillator problems are numerically implemented by sampling the path space for a given action by using Monte Carlo simulations. The average path converges to the solution of the equation of motion from classical mechanics, analogously as a canonical system is sampled for a given energy by computing the average state, finding the least energy state. Thus, this procedure can be general enough to solve other differential equations in physics and a useful tool to calculate the time-dependent properties of dynamical systems in order to understand the non-equilibrium behavior of statistical mechanical systems.

## 1. Introduction

The main objective of this work is to show a new computational framework based on Monte Carlo Metropolis, for the study of dynamical systems that are described by a Lagrangian, being a first approach for the understanding of non-equilibrium statistical mechanics (NESM) by using constraints, as performed in statistical mechanics.

Here, we propose a technique capable of simulating deterministic, dynamical systems through a stochastic formulation. This technique is based on sampling a statistical ensemble of paths defined as having the maximum path entropy (also known as the caliber) available under imposed time-dependent constraints. In Section 2, we introduce this approach, which is known as the maximum caliber principle (MaxCal) [1], a generalization of Jaynes’ principle of maximum entropy [2,3,4].

The principle of maximum entropy (MaxEnt) is a systematic method for constructing the simplest, most unbiased probability distribution function under given constraints, a conceptual generalization of Gibbs’ method of ensembles in statistical mechanics. The complete generality of the principle makes it widely used in several areas of science, such as astronomy, ecology, biology, quantitative finance, image processing, electronics, and physics, among others. According to Jaynes, choosing a candidate probability distribution by maximizing its entropy is a rule of inferential reasoning far beyond its original application in physics, which makes this rule a powerful tool for creating models in any context.

Monte Carlo Metropolis (MCM) is a rejection technique that can be used for sampling any probability distribution [5,6]. This probability distribution is usually constructed by using MaxEnt and optimized using the simulated annealing algorithm [7]. MCM and simulated annealing are used for the understanding of different systems in physics [8,9] such as spin models, material simulations, and thermodynamics, among others. In Section 3, an analogous methodology is presented, by assigning probabilities to paths instead of probabilities of states, allowing the minimization of functional quantities such as the classical action instead of the energy.

It is possible to directly use MaxEnt instead of MaxCal as an inference methodology for dynamical systems by constraining time-dependent functions, and this has been used to understand Newtonian dynamics [10] and the Schrödinger equation [11] in terms of traditional MaxEnt. A new approach has been explored recently for recovering frameworks for dynamical systems by using MaxCal and the path space [12,13,14,15]. In Section 4 and Section 5, we show a numerical implementation of this new approach, exposing the capability for solving complex problems in NESM. In summary, this constitutes a novel approach to the study of dynamical systems via Monte Carlo simulation.

## 2. Creating a Path Ensemble: The Maximum Caliber Principle

The maximum caliber principle allows the definition of a unique path ensemble given prior information and a number of dynamical constraints [1,14,16].

MaxCal is similar to the maximum entropy principle (MaxEnt) [13,15,17,18,19], but defined over the path space, allowing defining a probability functional P[x] as follows. Consider an *N*-dimensional path *x* on the path space X. In order to construct a probability functional for each path P[x], given an initial probability (prior) $P_{0}\left[x\right]$ and an arbitrary constraint:(1)$$\langle A[x]\rangle =a,$$
the caliber (or path entropy):(2)$$S\left[P_{0}\to P\right]=-\int_{X}D_{x}P\left[x\right]\ln\frac{P[x]}{P_{0}\left[x\right]},$$
must be maximized under the constraint in Equation (Equation 1) and the requirement that probability be normalized,
$$\int_{X}D_{x}P\left[x\right]=1.$$

The operation $\int_{X}D_{x}$ is a path integral, where $D_{x}$ is a differential in the path space X. Then, the probability functional obtained is,
(3)$$P\left[x|\beta \right]=\frac{1}{Z(\beta )}P_{0}\left[x\right]\exp\left(-\beta A\left[x\right]\right),$$
where Z(β) is the partition function and β is the Lagrange multiplier, given by the constraint equation:(4)$$-\frac{\partial }{\partial \beta }\ln Z\left(\beta \right)=a.$$

In this formalism, analogous to Gibbs’ method, β is identified as the inverse of the temperature ($\beta =1/k_{B}T$) similar to the canonical ensemble of statistical mechanics. Here, the expected value of an arbitrary functional F[x] is given by:(5)$${\langle F[x]\rangle }_{\beta }=\frac{1}{Z(\beta )}\int_{X}D_{x}\exp\left(-\beta A\left[x\right]\right)\;F\left[x\right],$$
but, perhaps more importantly, the expectation value of a function over time can be defined similarly as:(6)$${\langle f(\dot{x},x,t)\rangle }_{\beta ,t}=\frac{1}{Z(\beta )}\int_{X}D_{x}\exp\left(-\beta A\left[x\right]\right)\;f\left(\dot{x},x,t\right).$$

This last relation shows that MaxCal can be used to understand the macroscopic properties of time-dependent systems, which are the main elements in NESM.

## 3. Least Action Principle and Most Probable Path

From classical mechanics, it is well known that the path followed by a particle under a potential V(x;t) under the boundary conditions x(0)=0 and $x\left(T\right)=x_{f}$ is the one given by the least action principle [20,21], which in practice leads to an equation of motion describing the evolution of the particle from zero to *T*.

The classical action is a functional defined as $A\left[x\right]=\int_{0}^{T}dt\;L\left(\dot{x},x;t\right)$, where $L(\dot{x},x;t)$ is called the Lagrangian of the system, which for classical systems is:$$L\left(\dot{x},x;t\right)=\frac{m{\dot{x}}^{2}}{2}-V\left(x;t\right).$$

For a MaxCal framework where the classical action is constrained, following an analogous treatment to the constraint in Equation (Equation 1), the probability functional is of the form given in Equation (Equation 3). The most probable path can be obtained by finding the extrema of the functional P[x], by solving the equation $\frac{\delta P[x]}{\delta x(t^{\prime })}=0$. This is because the exponential function in Equation (Equation 3) is convex and monotonically increasing, and so, maximum probability is equivalent to imposing that *x* should be an extremum for the argument of the exponential,
(7)$$-\beta \;\frac{\delta A[x]}{\delta x(t^{\prime })}=0.$$

If the Lagrange multiplier is positive (β>0), the requirement is that the action is actually a minimum. This equation is precisely the Euler–Lagrange equation of motion for the Lagrangian [20,22]. In summary, the most probable path and the least action path in a MaxCal framework are the same. According to this, sampling trajectories from the probability distribution in Equation (Equation 3) using Monte Carlo methods and computing the averages of quantities should converge to a description of the dynamical properties of a classical system evolving in time.

Another important consequence of the use of MaxCal and the form of the probability is that the most probable path in general coincides with the average path according to the central limit theorem.

## 4. Computational Method

In order to define the elements on the path space X, for an *N*-dimensional path *x*, it is always possible to write it in an orthonormal basis $\phi_{i}$ of the form:(8)$$x\left(t\right)=\sum_{i}^{N}a_{i}\phi_{i}\left(t\right)=x\left(t;a\right).$$

Then, by changing the parameters $a_{i}$, it is possible to map the entire space of paths *x*. In other words, there is a one-to-one correspondence between an arbitrary path *x* and its parameter vector a, so the action becomes a function of a, namely A(a):=A[x(·;a)], and the problem of path sampling reduces to ordinary sampling [23] of *N*-dimensional states a,
(9)$$P\left(a|\lambda \right)=\frac{1}{Z(\lambda )}\exp\left(-\lambda A\left(a\right)\right).$$

The choice of the basis functions $\phi_{i}$ is, in principle, arbitrary. However, a convenient choice can be made related to the particular conditions of the problem to be solved. In this case, the target is the study of classical dynamical systems by using the MCM where most of the problems have well-defined boundary conditions; therefore, it is important to find a basis set in which one can easily generate paths in the desired path space holding the required, fixed boundary conditions. For this reason, we considered the Bézier curves as a basis set.

Bézier curves are defined by “control points” $c_{i}$, where the first $c_{0}$ and the last $c_{n}$ control point determine the boundary conditions of the curve, allowing the mapping of the path space with well-defined boundary conditions.

For an *N*-dimensional path *x* with boundary conditions $x\left(t_{0}\right)=c_{0}$ and $x\left(t_{f}\right)=c_{n}$, a Bézier curve is defined of the form:(10)$$x\left(t\right)=\sum_{i=0}^{N}c_{i}\;B_{i}\left(t;n\right),$$
where the basis functions $B_{i}\left(t;n\right)$ are the Bernstein polynomials,
(11)Bi(t;n)=ni(t−t0)i(tf−t)n−i(tf−t0)n.

Following these definitions, it is clear that all Bézier curves automatically follow the specified boundary conditions at $t=t_{0}$ and $t=t_{f}$.

## 5. Implementation of the Monte Carlo Metropolis for the Sampling Path Space

The Monte Carlo Metropolis (MCM) implementation is usually employed for sampling a multidimensional space governed by a probability distribution [5,6]. The following MCM implementation is used for sampling the path space X governed by a probability functional obtained via MaxCal. This procedure makes it possible to find the minimum action path.

For a given action A[x] and boundary conditions $x\left(t_{0}\right)=c_{0}$ and $x\left(t_{f}\right)=c_{n}$, the MCM evolution for sampling the path space is implemented as shown in Figure 1.

By performing this process in an iterative way, the path space is sampled, allowing the calculation of the properties for the system which is determined by the classical action used. As shown in Section 3, the probability distribution obtained when the classical action is constrained allows the sampling of the path space where the most probable path and the least action path coincide. Finally, λ can be understood as an analogue of the inverse of temperature in the usual MCM, due to the fact that, as λ→0, the sampled paths will be randomly more spread over the space, while taking λ→∞ constrains the sampled paths closer to the least action path. The value for λ in an MCM is related to the change from *x* to $x^{\prime }$, and empirically, this change must be adjusted to have approximately a 80% acceptance rate. We used the value λ=100 for all the MCM simulations performed in this work.

## 6. Results and Discussion

### 6.1. Free Particle Action

The equation of motion for a free particle is obtained by minimizing the classical action:$$A\left[x\right]=\int_{t_{0}}^{t_{f}}dt\;\frac{m\dot{x}{\left(t\right)}^{2}}{2}$$
according to the least action principle. For a free particle with mass m=1 and boundary conditions x(0)=0 and x(1)=1, the analytical solution for the least action path is the straight line x(t)=t.

By using MCM simulation as shown in Figure 2, we sampled the path space and calculated the simple average position $\bar{x}\left(t\right)$ at each time. We see that the simulation readily converges to the correct least action path.

For this simulation, five control points were used and were sufficient to obtain the exact result ($R^{2}=0.99$) in less than 10,000 Monte Carlo steps.

### 6.2. Harmonic Oscillator Action

In the case of the harmonic oscillator, the action is of the form:(12)$$A\left[x\right]=\int_{t_{0}}^{t_{f}}dt\;\left(\frac{m\dot{x}{\left(t\right)}^{2}}{2}-\frac{kx{\left(t\right)}^{2}}{2}\right).$$

Without loss of generality, for numerical simulations, m=1 and k=1 were used. The solution for this problem was divided into two parts. The first solution found was the least action path for a short time interval. More precisely, we simulated a particle with boundary conditions x(0)=0 and $x\left(t_{f}\right)=a\sin\left(\omega t_{f}\right)$, with $t_{f}$ less than the half period $\frac{T}{2}$.

In this case, the least action path also converges to the analytical solution, correctly solving the equation of motion as shown in Figure 3. For this simulation, five control points were used and were sufficient to obtain the exact result ($R^{2}=0.99$) in less than 10,000 Monte Carlo steps.

The second case corresponds to the harmonic oscillator with boundary conditions x(0)=0 and $x\left(t_{f}\right)=a\sin\left(\omega t_{f}\right)$, but where $t_{f}$ is larger than the half period $\frac{T}{2}$. In other words, the end condition is past the first node of the harmonic oscillator. Under these boundary conditions, an unexpected result is obtained: the Monte Carlo sampling procedure diverges. This result can be understood as due to the fact that, for this case, the action extremum is not a global minimum [24]. More precisely, the second functional derivative for the action of the harmonic oscillator action shows that the extremum is a saddle point in the case where the total time is longer than half a period, and a “true” minimum only for paths with total time less than half a period. We solved this convergence problem by considering an additional constraint to the action solved, suggested by the work of Gray and Taylor [24] in classical mechanics. The constraint involves the so-called kinetic foci, defined by the condition:(13)$$\frac{\partial x}{\partial v_{0}}=0.$$
where $v_{0}$ is the initial velocity for the path *x*. As it turns out, the action extremum is guaranteed to be a minimum if the paths used pass close enough to a kinetic focus $x_{i}$. This can be implemented in the Monte Carlo simulation by including a quadratic constraint in the probability functional, leading to:(14)$$P\left[x|\lambda ,\beta \right]=\frac{1}{\eta (\lambda ,\beta )}P_{0}\left[x\right]\exp\left(-\lambda A\left[x\right]-\beta \sum_{i}{\left(x\left(t_{i}\right)-x_{i}\right)}^{2}\right),$$
where $\left(t_{i},x_{i}\right)$ are the set of kinetic foci (solutions of Equation (Equation 13)) and β>>λ, in order to stop the system from drifting away from the action extremum.

As an example, we solved the harmonic oscillator for a total time close to one and a half periods $\frac{3T}{2}$, sampling all the paths that cross the first two kinetic foci of the harmonic oscillator, namely the points $\left\{\left(0,\frac{T}{2}\right),\left(0,T\right)\right\}$. Now, the Monte Carlo procedure does converge to the expected solution as shown in Figure 4.

For this simulation, eight control points were used and were sufficient to reach the exact result ($R^{2}=0.95$) in less than 20,000 Monte Carlo steps. An important remark here is to note the number of Bézier basis elements, or control points, because this is closely related to the computing time. For this reason, the main goal in an efficient simulation is to find the minimum number of control points to use without sacrificing precision, needed to map any solution of the differential equation of interest.

## 7. Concluding Remarks

In summary, we described a technique for implementing Monte Carlo sampling of dynamical trajectories in classical Lagrangian systems under the maximum caliber formalism. We demonstrated its usefulness by applying this technique to the case of the classical free particle and harmonic oscillator, recovering in both cases a statistical distribution of paths centered on the classical solution of the Euler–Lagrange equation. For the case of the harmonic oscillator, we noted the need for fixing additional points known as the kinetic foci of the system in order for the simulation to converge properly. Our proof-of-concept implementation could be the starting point for a complete computational scheme of the simulation of classical systems under uncertainty. It remains to be seen how this method scales to multidimensional and many-particle interacting systems. This framework could be used for the study of dissipative systems with a known Lagrangian [25], being a promising line of work for the study of non-equilibrium physical problems. Finally, one of the main proposed uses of this framework is to obtain the instantaneous probability density of positions at each time, which would allow obtaining the instantaneous macroscopic properties of classical systems under uncertainty [26].

## Figures and Tables

**Figure 1 entropy-22-00916-f001:**
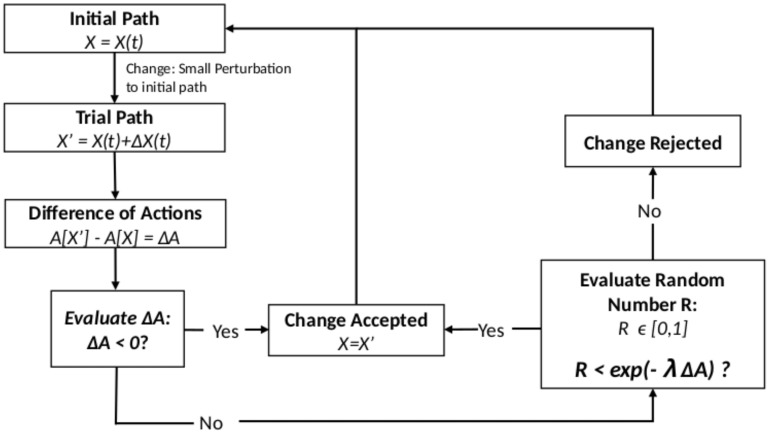
Explanatory diagram for a Monte Carlo Metropolis (MCM) sampling in the path space.

**Figure 2 entropy-22-00916-f002:**
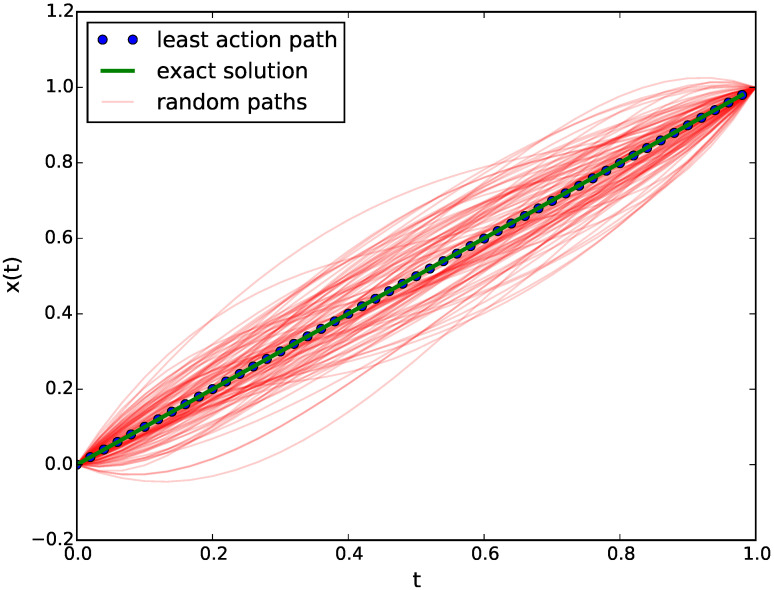
Dynamical trajectories sampled for the free particle action.

**Figure 3 entropy-22-00916-f003:**
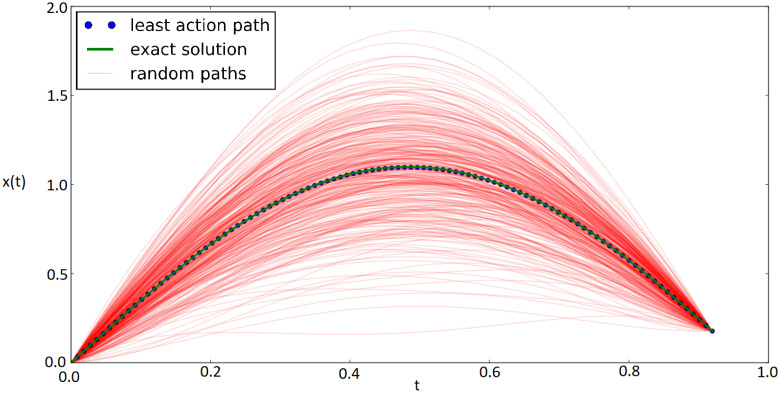
Paths sampled for the harmonic oscillator action considering short time intervals.

**Figure 4 entropy-22-00916-f004:**
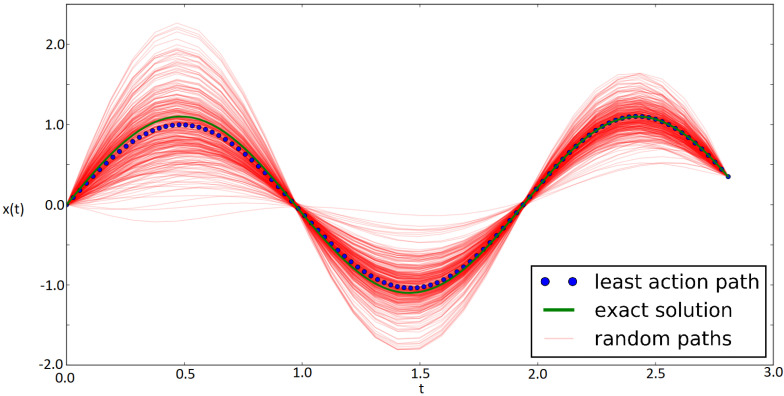
Harmonic oscillator with fixed kinetic foci.

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
