# Peer review of "Solving Equations of Motion by Using Monte Carlo Metropolis: Novel Method Via Random Paths Sampling and the Maximum Caliber Principle"

_entropy, 2020, doi:10.3390/e22090916_

Round 1

Reviewer 1 Report

The manuscript "Solving equations of motion by using Monte Carlo Metropolis ..." has an interesting application of a version of the Metropolis algorithm to dynamical systems. The paper needs a bit of rewriting to clarify the ideas. Listed below are some queries and some suggestions.

1. The first sentence of the Introduction needs to be rewritten to make it clear that the approach to solving non-equilibrium statistical mechanics problems will involve a Monte Carlo method.

2. Page 2, 4th paragraph, 2nd sentence. The authors are confusing the Metropolis algorithm with one of its very famous applications, Simulated Annealing. The Metropolis algorithm is a rejection technique that can be used to sample any distribution function. Standard descriptions can be found in W.W. Wood and J.J. Erpenbeck, Annual Review of Physical Chemistry, <27>, 319, 1976 or M.P. Allen and D.J. Tildesley, Computer Simulations of Liquids, Oxfod University Press, 1993. In S. Kirkpatick, C.D. Gelattt and M.P. Vecchi, Science, <220>, 671 ,1983, the idea of using an entropy measure as analogous to the Gibb's measure was introduced to solve an optimization problem. The current authors propose a similar idea in their Eqns. (3) and (4). A more general description of Simulated Annealing is given in P.J. van Laarhoven and E.H. Aarts, Simulated Annealing: Theory and Applications, Spring-Verlag, 1987.

3. Page 2, 5th paragraph, 1st sentence. The abbreviation MaxCal is first used here without a definition, i.e. Maximum Caliber. The first sentence needs to be rewritten since the previous two paragraphs were discussing MaxEnt, not MaxCal.

4. Page 3, 2nd paragraph of Section II, third line. The symbol X (upper case) is introduced without a definition. The definition is given on page 5, in the first line of Section IV.

5. In Eq. (2), the symbol Dx need to be defined.

6. After Eq.(4), it is the analogy to the Gibb's measure that allows β to be identified as the inverse temperature.

7. Page 5, the sentence before Eq.(9). The authors state the "... the problem of path sampling reduces to ordinary sampling of N-dimensional states a". This needs a better explanation and a reference.

8. Page 6, Section V, first sentence. A reference to the standard description of the Metropolis algorithm is missing. Either of the two references mentioned in 2. above or the manuscript's reference 5 could be used.

9. Page 6, Section V, 3rd paragraph, 3rd sentence. The identification of λ with the inverse temperature is a characteristic of Simulated Annealing not the more general Metropolis algorithm. The authors mention that the variation of λ as the simulation continues is an empirical process. That is true and the process is often called the "cooling schedule". A description and justification for the choice of λ is needed in each of the examples given.

10. Page 7, Section VI, subsection A. In the second paragraph, the authors state that the simulation converges readily to the correct path. A description of how λ was changed as the simulation proceeded is needed. If λ did not need to be changed, that should be stated.

11. Page 7, Section VI, subsection A, paragraph 3. Giving an average timing for convergence does not mean anything without more information. To give a correct perspective, the authors need to mention the CPU chip that was used, how the pseudorandom numbers were generated, and the computational environment. Since the timing of the computation is not crucial to the author's example, perhaps it should just be left out.

12. Page 7, Section VI, subsection B. Either give more information or leave out the timings. The variation of λ as the simulation proceeds needs to be described.

13. Page 8. In Eq.(13), the variable v0 needs to be defined.

14. Page 9. The authors mention that the computing time is closely related to the number of Bézier basis elements. Does this imply that a different set of basis functions would improve the computing time? Did the authors consider alternative basis functions?

Author Response

As suggested, points 1, 2, 3, 4, 5, 6, 8, 11, 12 and 13 were implemented.

Regarding the points 9 and 10, it was not necessary to implement the simulated annealing methodin order to find the least action path. The methodology used was a Monte Carlo Metropolis atfixed low temperature. In this work the value λ= 100was used.

Regarding point 14, the computing time can be improved using a smaller number of control pointswhen the Bézier basis is used. It is possible to use another basis for the problem (as, for instance, aFourier basis) but according to our results the Bézier basis is optimal. We consider the use of theFourier basis for the problem, but it does not perform well for fixed-boundary-condition paths foreach Monte Carlo step.

Reviewer 2 Report

enclosed attachement

Author Response

As suggested, we changed “partial differential equations” to “Euler-Lagrange equations” in theabstract. Certainly, MCM and MaxEnt are well known methods but the procedure to solve theEuler-Lagrange equation by using MCM is not common and represents the original contribution ofthis work. We would like to thank the referee for his/her straightforward suggestion that allows usimproving the abstract and introduction. The symbol at the end of the first sentence in Section Vwas corrected.

Reviewer 3 Report

The results presented in the manuscript are very interesting and may be useful for the researchers on this subject. The authors use a numerical procedure based on Monte Carlo simulations to analyze classical Lagrangian systems. In my opinion, this article deserves to be considered for publication. However, before getting the publication, the authors should discuss in more detail (a few words) the situations characterized by nonquadratic Lagrangian systems or systems with dissipative terms. 

Author Response

We include into conclusions a comment remarked by the referee, related to the fact that theframework proposed is not restricted to conservative systems, and actually the implementation fornonquadratic Lagrangian systems is an idea possible to explore. When using another Lagrangian,the kinetic foci should be considered, which in principle could be obtained numerically oranalytically depending on each problem.

Round 2

Reviewer 2 Report

The authors answered satisfactory my questions and comments